# Changes in Sleep Problems in Patients Who Underwent Surgical Treatment for Degenerative Spinal Disease with a Concurrent Sleep Disorder: A Nationwide Cohort Study in 3183 Patients during a Two-Year Perioperative Period

**DOI:** 10.3390/jcm11247402

**Published:** 2022-12-14

**Authors:** Jihye Kim, Jang Hyun Kim, Tae-Hwan Kim

**Affiliations:** 1Division of Infection, Department of Pediatrics, Kangdong Sacred Heart Hospital, Hallym University College of Medicine, 150 Seongan-ro, Gangdong-gu, Seoul 05355, Republic of Korea; 2Spine Center, Department of Orthopedics, Hallym University Sacred Heart Hospital, Hallym University College of Medicine, 22 Gwanpyeong-ro, 170beon-gil, Dongan-gu, Anyang 14068, Republic of Korea

**Keywords:** change, sleep disturbance, sleep medication, sleep improvement, surgery, degenerative spinal disease, sleep disorder

## Abstract

Sleep disturbance is prevalent in patients with degenerative spinal disease, and recent studies have reported that surgical treatment is more effective for improving sleep quality than conservative treatment. We aimed to investigate the perioperative changes of sleep problems in patients who underwent surgical treatment for degenerative spinal disease with a concurrent sleep disorder, and presented them according to various clinical profiles possibly associated with sleep disturbance. In addition, we identified factors associated with poor sleep improvement after surgery. This study used data from the Korea Health Insurance Review and Assessment Service database from 2016 to 2018. We included 3183 patients aged ≥19 years who underwent surgery for degenerative spinal disease and had a concurrent sleep disorder. Perioperative changes in the two target outcomes, including the use of sleep medication and hospital visits owing to sleep disorders, were precisely investigated according to factors known to be associated with sleep disturbance, including demographics, comorbidities, and spinal regions. Logistic regression analysis was performed to identify factors associated with poor improvement in terms of sleep medication after surgery. All estimates were validated using bootstrap sampling. During the 1-year preoperative period, the use of sleep medications and hospital visits owing to sleep disorder increased continuously. However, they abruptly decreased shortly after surgical treatment, and throughout the 1-year postoperative period, they remained lower than those in the late preoperative period. At the 1-year follow-up, 75.6% (2407 of 3183) of our cohort showed improvement in sleep medication after surgery. Multivariable analysis identified only two variables as significant factors associated with non-improvement in sleep medication after surgery: depressive disorder (odds ratio (OR) = 1.25 [1.06–1.48]; *p* = 0.008), and migraine (OR = 1.42 [1.04–1.94]; *p* = 0.028). We could not investigate the actual sleep quality and resultant quality of life; however, our results justify the necessity for further high-quality studies that include such information and would arouse clinicians’ attention to the importance of sleep disturbance in patients with degenerative spinal disease.

## 1. Introduction

Despite the importance of sleep for physiologic function [1,2,3,4], falling asleep is not easy for modern people, and sleep disturbance is prevalent in the general population [5,6,7]. The importance of sleep has been particularly emphasized for older patients with chronic medical conditions because they frequently experience sleep disturbance which significantly influences their disease outcome and life expectancy [4,8,9,10,11,12]. Therefore, sleep is an important topic for clinicians. Sleep has not been a subject of interest for researchers and clinicians caring for patients with spinal disease. However, studies have begun to address the fact that sleep disturbance is also highly prevalent in these patient populations [13,14,15,16,17], suggesting various mechanisms for their sleep disturbance [13,14].

Recently, two single-center prospective studies reported changes in patients’ sleep quality after surgery for degenerative spinal disease and compared these results with those experienced by patients after conservative treatment [18,19]. They concluded that surgical treatment for degenerative spinal disease improves sleep quality more effectively than conservative treatment, especially after a long-term follow-up. Degenerative spinal disease is a common medical problem globally [20], and 11 to 74% of these diseases are accompanied by sleep disturbance [13,14,15,17,18,19,21]. Therefore, the results of the two prospective studies are promising for numerous patients with degenerative spinal disease and chronic sleep disorder.

However, the limitations of these two studies should be addressed. First, these comparative studies were single-center studies with small sample sizes. The studies focused on a narrow range of patients with specific types of degenerative spinal disease in a specific spinal region, such as lumbar stenosis or cervical myelopathy. Therefore, they could not comprehensively analyze various types of degenerative spinal disease, which include the entire spinal region. In addition, because the primary purpose of the two studies was to compare sleep improvement between the conservative and surgical groups, they did not require a large number of patients who underwent surgical treatment. Therefore, the number of surgical cohorts was small (<50 patients), and multivariable analysis to identify factors associated with adverse sleep outcomes after surgery could not be performed because of the limited sample size. Second, patients’ sleep disturbance was identified using only single-time questionnaire-based surveys, which could result in the overestimation of sleep problems [18,19]. Therefore, there was a higher chance of including patients with temporary or minor sleep problems.

We planned a nationwide, population-based cohort study using a claims database. This study had two research purposes. First, we investigated changes in sleep problems in patients who underwent surgical treatment for degenerative spinal disease and had a concurrent sleep disorder. The use of sleep medication and hospital visits owing to sleep disorders were chosen as target outcomes in our study. This is because the claims database guarantees the reliability and accuracy of these two pieces of information as important objective measures of sleep disturbance [22,23]. In addition, perioperative changes in the two target outcomes were thoroughly presented according to the various clinical profiles possibly associated with sleep disturbance, including demographics, comorbidities, and spinal regions. Second, we identified the factors associated with poor sleep improvement after surgery for degenerative spinal disease.

## 2. Materials and Methods

### 2.1. Database

The data were obtained from the Health Insurance Review and Assessment Service (HIRA) database which contains data from all the hospitals and community clinics in Korea. In the database, the seventh revision of the Korean Classification of Diseases and tenth revision of the International Classification of Diseases (ICD-10) were used to identify diseases. The anatomical therapeutic chemical (ATC) codes and HIRA general name codes were used to identify the drugs used. In addition, this study was approved by the Institutional Review Board of our hospital (IRB No. 2020-03-009-001).

### 2.2. Study Patients

Patients aged >19 years who underwent surgical treatment (index surgery) for degenerative spinal disease from 1 January 2016 to 31 December 2018, were included (Figure 1). We identified degenerative spinal diseases using the following ICD-10 codes: spinal stenosis (M48.0), spondylolisthesis (M43.1), spondylolysis (M43.0), other spondylosis (M47.1 and M47.2), and cervical disc disorder (M50). Information regarding spinal surgery and its regions were identified using the corresponding electronic data interchange codes (Appendix A). Subsequently, we excluded patients with spinal infection, spinal fractures, or malignancy within 2 years before the index surgery (Figure 1). The ICD-10 codes for exclusion are presented in Appendix A.

Sleep disorder of the cohort patients was initially identified using the following ICD-10 codes: nonorganic sleep disorders (F51) and sleep disorders (G47). To identify patients with a concurrent sleep disorder, we only included patients who visited the hospital with the two ICD-10th codes as the primary diagnosis of the visits within 1 year before the index surgery.

### 2.3. Two Target Outcomes Associated with Sleep Disturbance: Use of Sleep Medication and Hospital Visits Owing to Sleep Disorders

Our main target outcome of interest was the use of sleep medication during the perioperative period. Sleep medication was defined as currently available drugs for insomnia approved by the Korean Food and Drug Administration [24]. They included flurazepam, triazolam, flunitrazepam, brotizolam, zolpidem, eszopiclone, doxepin, doxylamine, and diphenhydramine. Among them, antihistamines such as doxylamine and diphenhydramine were excluded. The sleep medication codes are presented in Appendix A. Based on this definition, we investigated both the types of the sleep medication and their prescribed duration during the perioperative period. In addition, we investigated the number of hospital visits with a sleep disorder as the primary diagnosis during the perioperative period.

### 2.4. Definition of the Eight Quarters of Perioperative Periods

To observe the changes in the two target outcomes, we defined the perioperative period as the period between 1 year before and after the index surgery. Subsequently, we divided each preoperative and postoperative period into four intervals (quarters), each lasting 90 days (Figure 2). The 1-year preoperative period was divided into the following four quarters: (1) within 90 days before the index surgery (preoperative 4th quarter); (2) between the 91st and 180th day before the index surgery (preoperative 3rd quarter); (3) between the 181st and 270th day before the index surgery (preoperative 2nd quarter); and (4) between the 271st and 360th day before the index surgery (preoperative 1st quarter). The 1-year postoperative period was also divided into the following four quarters: (1) within 90 days after the index surgery (postoperative 1st quarter); (2) between the 91st and 180th day after the index surgery (postoperative 2nd quarter); (3) between the 181st and 270th day after the index surgery (postoperative 3rd quarter); and (4) between the 271st and 360th day before the index surgery (postoperative 4th quarter).

### 2.5. Evaluation of Perioperative Changes in the Two Target Outcomes

We investigated the use of sleep medication and hospital visits owing to sleep disorders during the perioperative period. The two target outcomes were thoroughly investigated and presented according to four groups of factors known to be associated with sleep disturbance in patients with degenerative spinal disease [21]: demographics; general medical comorbidities; neuropsychiatric disorders; and concurrent osteoarthritis of the extremities.

Information on the four groups of factors was retrieved using the previously described method [21]. Demographic data at the time of index surgery, including age, sex, region of residence, and hospital type, were initially retrieved. General medical comorbidities and neuropsychiatric disorders diagnosed within 1 year before the index surgery were investigated using the corresponding ICD-10 codes (Appendix A), and the Charlson comorbidity index (CCI) score was calculated using an optimized method for the database using ICD-10 codes [25,26,27,28]. Diagnosis of depression was confirmed by the use of antidepressants (ATC code: N06A, Appendix A). Osteoarthritis in the extremities was identified using the validated method for our database: [29] (1) ICD-10 codes for osteoarthritis (M15 to M19), and (2) corresponding X-rays of the extremities. The x-ray codes are presented in Appendix A.

### 2.6. Statistical Analysis

During eight quarters of the perioperative period, changes in the use of sleep medication and number of hospital visits owing to sleep disorder were precisely presented according to the four groups of factors known to be associated with sleep disturbance in patients with degenerative spinal disease [21].

Expecting an overall postoperative decrease (improvement) in the use of sleep medication, [18,19] we tried to analyze factors associated with poor sleep improvement regarding the use of sleep medication. To assess the changes in sleep medication at 1-year follow-up after surgery, we compared the prescribed duration of the sleep medication between the preoperative and postoperative 4th quarters (Figure 2) and identified patients who showed no improvement in the duration of sleep medication between the two periods. Subsequently, we calculated the proportions of patients (%) who showed no postoperative improvement in the duration of sleep medication according to the four groups of factors associated with sleep disorder. The statistical difference of the proportions was presented with a standardized mean difference (SMD) known as Cohen’s *d.* Logistic regression analysis was performed to identify independent factors associated with patients who showed no postoperative improvement in the use of sleep medication. Adjustments were made for variables identified as significant in the univariable analysis (SMD > 0.1).

The statistical model was internally validated using the bootstrap method. All estimates in the multivariable model were presented with relative bias estimated on 1000 bootstrapped samples. Relative bias was defined as the difference between the mean bootstrapped regression coefficient estimates and the mean parameter estimates of the multivariable model divided by the mean parameter estimates of the multivariable model. In addition, the variance inflation factor was used to evaluate multicollinearity between covariates. Data extraction and statistical analysis were performed using the SAS Enterprise Guide 6.1 (SAS Institute, Cary, NC, USA).

## 3. Results

Among the 198,844 patients who underwent spinal surgeries (index surgery) for degenerative spinal diseases from 2016 to 2018, we excluded patients who had malignancy (*n* = 11,504), spinal infection (*n* = 1937), spinal fracture (*n* = 81,463) within 2 years before the index surgery, and those with missing data (*n* = 376, Figure 1). Among the remaining 106,837 patients, 3148 (2.9%) who visited the hospital because of sleep disorders were included in our study. Baseline information regarding the cohort patients is presented in Table 1. Their mean age was 66.7 years, and 59% (*n* = 1875) were women.

### 3.1. Overall Perioperative Changes in the Two Target Outcomes

In the entire cohort, both the use of sleep medication and hospital visits owing to sleep disorders increased continuously throughout the preoperative period (Figure 3). The rate of this increase also increased continuously during the preoperative period, which was highest immediately before the index surgery, in the preoperative 4th quarter. However, shortly after the index surgery, the use of sleep medication and hospital visits owing to sleep disorders decreased abruptly. The rate of this decrease was highest in the postoperative 1st quarter. There were fluctuations in the subsequent postoperative period; however, the postoperative use of sleep medication and hospital visits were persistently lower than those in the preoperative 4th quarter. At the 1-year follow-up, 24.4% (776 of 3183) of patients in the cohort showed improvement in the prescribed duration of sleep medication (Table 1).

The two most commonly used sleep medications were zolpidem and triazolam (Figure 4). The proportions of these two drugs increased continuously during the preoperative period and decreased during the subsequent postoperative period. Except for these two drugs, there were no remarkable changes in the use of other drugs.

### 3.2. Perioperative Changes in the Two Target Outcomes: According to the Demographics and Spinal Regions

Hospital visits owing to sleep disorders decreased after surgery regardless of age group and sex; however, the postoperative decrease in the use of sleep medication was more pronounced in the female and older age groups (Figure 5).

Perioperative changes in the use of sleep medication and hospital visits owing to sleep disorders are presented according to the spinal regions in Figure 6. The patterns of perioperative changes in the cervical and lumbar regions were similar to those in the entire cohort. In contrast, the preoperative pattern in the use of sleep medication and hospital visits owing to sleep disorders in patients with thoracic lesions differed from the overall pattern shown in Figure 3. The use of medication and hospital visits in patients with thoracic lesions decreased continuously, even during the preoperative period, although the most abrupt decrease occurred after surgery.

### 3.3. Perioperative Changes in the Two Target Outcomes: According to the Comorbidities

Information about the comorbidities of the cohort patients is presented in Table 2. We also presented the changes in the use of sleep medication and hospital visits owing to sleep disorders according to the CCI scores (Figure 7), neuropsychiatric diseases (Figure 8), and concurrent osteoarthritis of the extremities (Figure 9).

Regardless of the type of comorbidity, the use of sleep medication and hospital visits owing to sleep disorders decreased abruptly in the postoperative 1st quarter, shortly after surgery. Subsequent postoperative use of sleep medication showed a rebound increase in the postoperative 2nd quarter; however, the use of sleep medication in the postoperative 4th quarter was lower than that in the preoperative 4th quarter in most patient subgroups (Figure 7, Figure 8 and Figure 9). On the other hand, in three patient subgroups, including those with CCI score ≥ 6 (Figure 7b), depression, and other-type headaches (Figure 8b), the mean use of sleep medication in the postoperative 4th quarter exceeded that in the preoperative 4th quarter.

### 3.4. Factors Associated with Poor Improvement in the Sleep Medication after Surgery: Internal Validation Using Bootstrap Sampling

At the 1-year follow-up after surgery, the duration of the sleep medication decreased (improved) in 75.6% of patients (2407 of 3183, Table 1). In the univariable analysis, various factors showed statistical significance by the standard of SMD > 0.1 (Table 1 and Table 2); however, the multivariable analysis identified only two variables as significant factors associated with non-improvement in sleep medication after surgery (Table 3): depressive disorder (odds ratio (OR), 95% confidence interval [CI] = 1.25 [1.06–1.48]; *p* = 0.008), and migraine (OR = 1.42 [1.04–1.94], *p* = 0.028). The relative bias of the estimates for most risk factors was low, except for moderate-to-severe liver disease (678.3%) and other-type headaches (−139.0%). Bootstrap-adjusted odds ratios and their 95% CIs for the multivariable model are presented in Table 3 and Figure 10.

## 4. Discussions

Chronic pain and its intensity are major risk factors for sleep disturbance [30,31,32,33]. However, overall pain intensity is not a significant risk factor for sleep disturbance in patients with degenerative spinal disease [13,14]. Instead, the radiologic degree of degeneration is a stronger predictor for sleep disturbance. For example, sleep disturbance in patients with lumbar stenosis was closely associated with the degree of foraminal-type stenosis [14], and sleep disturbance in patients with cervical myelopathy was associated with the degree of central-type stenosis [13]. If severe radiologic degeneration is a major component of sleep disturbance in patients with degenerative spinal disease, sleep disturbance will be prevalent in patients who undergo surgical treatment for degenerative spinal disease [14,18,19,21], and sleep disturbance might have influenced their choice of surgical treatment. In addition, we can reasonably expect that ‘decompressive’ surgical treatment for degenerative spinal disease will positively influence sleep quality. These hypotheses are clinically important and should be verified because sleep disturbance in patients with a severe degree of degenerative spinal disease is not easily improved by conservative treatment [18,19].

In this regard, we attempted to clearly visualize perioperative changes in sleep medication and hospital visits owing to sleep disorders in patients who underwent surgery for degenerative spinal disease with concurrent sleep disorders. During the preoperative period, the use of sleep medication and hospital visits owing to sleep disorders increased continuously (Figure 3). However, they abruptly decreased shortly after surgical treatment. In addition, throughout the 1-year postoperative period, they remained lower than those in the preoperative 4th quarter. When the use of sleep medication immediately before surgery (in the preoperative 4th quarter) was compared with that of 1 year after surgery (in the postoperative 4th quarter), 75.6% (2407 of 3183) of the patients in our cohort showed improvement in sleep medication after surgery. Multivariable analysis identified that the two neuropsychiatric disorders, depressive disorder and migraine, were associated with poor outcomes in the use of sleep medication after surgery.

One of the major advantages of our study is that we obtained accurate information about the two target outcomes during the 1-year preoperative period in all cohorts. As a claims database based on an obligatory national health insurance system, the HIRA database contains all inpatient and outpatient data for the entire population. In our claims database, information about the two target outcomes, including sleep medication and hospital visits owing to sleep disorders, has been prospectively recorded and reviewed by government officials; thus, it is available for all patients, regardless of the time interval. Investigating sleep problems during the 1-year preoperative period in all the study cohorts is theoretically impossible, even in randomized or prospective studies. This is because researchers would defer surgical treatment for degenerative spinal disease for 1 year without exception to prospectively collect such information during the 1-year preoperative period.

Therefore, one of the most important findings of our study is the changes in the two target outcomes during the year before the index surgery. During the preoperative period, the use of sleep medication and hospital visits owing to sleep disorders increased continuously. The increasing rates continued during the 1-year preoperative period, culminating in the preoperative 4th (last) quarter (Figure 3). However, the accelerated preoperative increase in the use of sleep medication and hospital visits immediately reversed after surgery and abruptly decreased in the postoperative 1st quarter. The decreasing rate at the postoperative 1st quarter was even faster than the increasing rate, which culminated in the preoperative 4th quarter (Figure 3).

In addition, this is the largest study investigating changes in sleep problems after surgical treatment for degenerative spinal disease and has several advantages. First, through a comprehensive analysis based on a sufficient number of patients (*n* = 3183), perioperative changes in the two target outcomes were clearly presented according to various factors possibly associated with sleep disturbance (Figure 5, Figure 6, Figure 7, Figure 8 and Figure 9). Second, we included enough patients with evident preoperative sleep disorders. In the two previous studies, sleep disturbance in the cohorts was identified using only single-time questionnaire-based surveys, with a high possibility of including patients with temporary or minor sleep problems despite their prospective study design. [18,19] However, we only included patients who visited the hospital with sleep disorders as the main reason for the hospital visit. In similar patient populations, the prevalence rates of sleep disturbance from large database studies conducted using ICD-10 codes (3.8 to 10.8%) are lower than those from survey-based single- or multi-center studies (42 to 74%) [13,14,15,16,17,21]. Therefore, our 3183 (2.9%) cohort patients, chosen using concordant ICD-10 codes for sleep disorder among 106,837 patients who underwent surgery for degenerative spinal disease, were expected to have more severe sleep disturbance than those chosen from questionnaire-based surveys [18,19] (Figure 1). They are more appropriate for evaluating the influence of spinal surgical treatment on sleep disturbance.

Accordingly, throughout the 1-year preoperative period, the mean duration of sleep medication use was approximately 30 days per quarter (90 days, Figure 3), which indirectly reflects their severe and chronic sleep disturbance. Nevertheless, their severe and chronic dependency on sleep medication and resultant hospital visits strikingly decreased after surgical treatment, regardless of various known risk factors for sleep disturbances (Figure 5, Figure 6, Figure 7, Figure 8 and Figure 9) [21]. After surgery, 75.6% (2407 of 3183) of the patients in our cohort with preoperative sleep disorders showed improvement in sleep medication. Among numerous independent variables, only two factors, depressive disorder and migraine, were significantly associated with poor outcomes regarding sleep medication in the multivariable analysis (Table 3 and Figure 10).

However, our study has limitations. First, our claims database was not originally designed for research. Therefore, important information possibly related to sleep disturbance, such as the anatomical degree of spinal degeneration or neurologic impairment, was not included. We can propose that most patients who underwent surgical treatment had severe degrees of degeneration; however, such additional information could have influenced our results [13,14]. To reduce the influence of such unknown confounders, we validated our results using bootstrap sampling, and the results were consistent. In addition, we used validated data-retrieving methods for the HIRA database, and thoroughly presented the exact diagnostic and therapeutic codes for all types of diseases and drugs to ensure the reproducibility of our results. Second, the two target outcomes, the use of sleep medication and hospital visits owing to sleep disorders, are not precise measures for sleep quality. Based on our results, we cannot assert that surgical treatment in patients with degenerative spinal disease and concurrent sleep disorders can improve actual sleep quality and the resultant quality of life. This conclusion is beyond the scope of our study. Instead, as a comprehensive preliminary study, our results justify the necessity of further high-quality studies investigating improvements in actual sleep quality after spine surgery in patients with degenerative spinal disease and concurrent sleep disorders.

In conclusion, our population-based study using a nationwide database clearly presented perioperative changes in the use of sleep medication and hospital visits owing to sleep disorders in patients who underwent surgery for degenerative spinal disease with a concurrent sleep disorder. During the 1-year preoperative period, the use of sleep medication and hospital visits owing to sleep disorder increased continuously. However, they abruptly decreased shortly after surgical treatment. In addition, throughout the 1-year postoperative period, they remained lower than those in the preoperative 4th quarter. At the 1-year follow-up, 75.6% (2407 of 3183) of patients in our cohort showed improvement in sleep medication after surgery. Multivariable analysis identified that the two neuropsychiatric disorders, depressive disorder and migraine, were associated with poor outcomes in the use of sleep medications after surgery. We could not investigate the actual sleep quality and resultant quality of life in our cohort; however, our results justify the necessity of further high-quality studies which include such information and would arouse clinicians’ attention to the importance of sleep disturbance in patients with degenerative spinal disease.

## Figures and Tables

**Figure 1 jcm-11-07402-f001:**
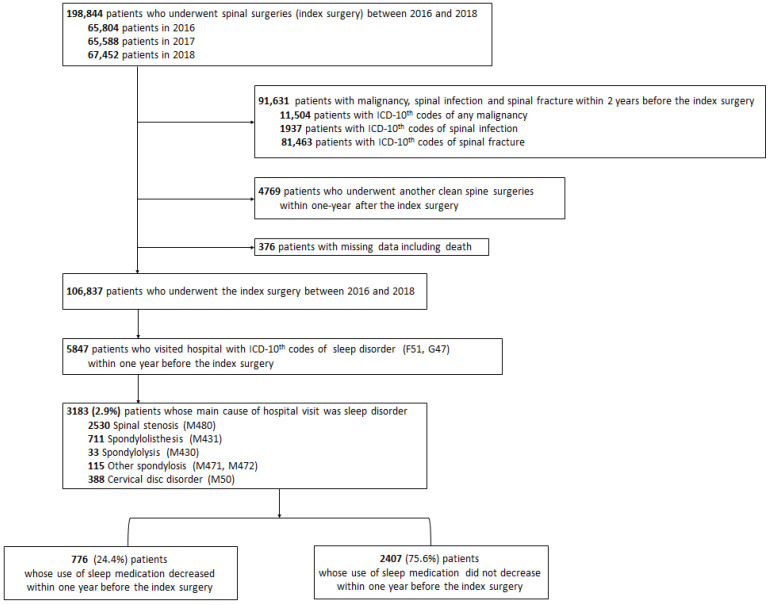
Patient enrollment.

**Figure 2 jcm-11-07402-f002:**
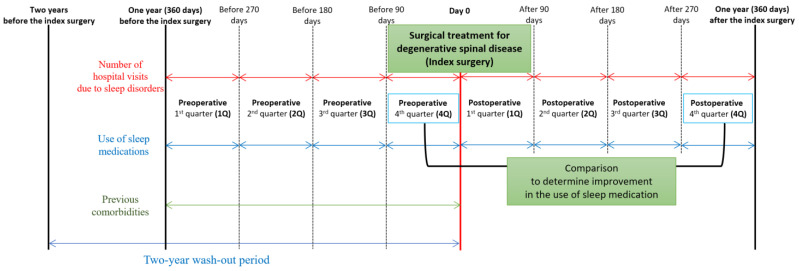
Study designs and definitions of the eight perioperative periods.

**Figure 3 jcm-11-07402-f003:**
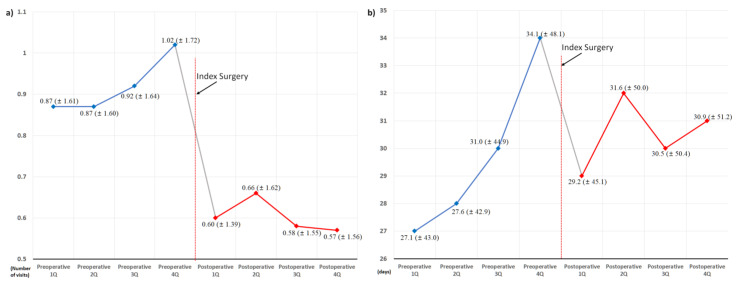
Overall perioperative changes in the use of sleep medication and hospital visits owing to sleep disorder: (**a**) changes in the number of hospital visits, and (**b**) changes in the prescribed duration of sleep medication.

**Figure 4 jcm-11-07402-f004:**
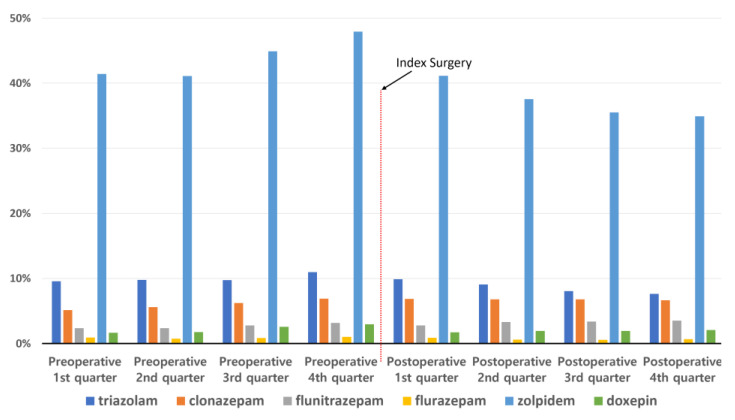
Types of sleep medication used during the perioperative period.

**Figure 5 jcm-11-07402-f005:**
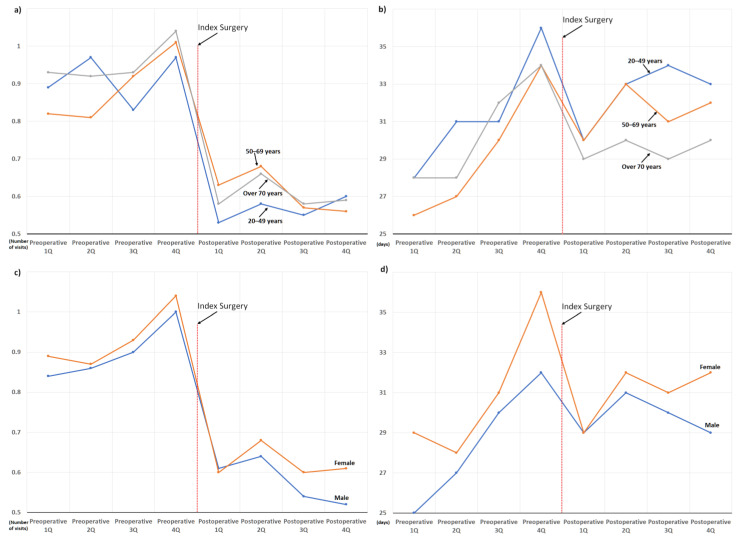
Perioperative changes in the use of sleep medication and hospital visits owing to sleep disorder according to the demographics: (**a**) changes in the number of hospital visits according to the age groups; (**b**) changes in the prescribed duration of sleep medication according to the age groups; (**c**) changes in the number of hospital visits according to sex groups; and (**d**) changes in the prescribed duration of sleep medication according to sex groups.

**Figure 6 jcm-11-07402-f006:**
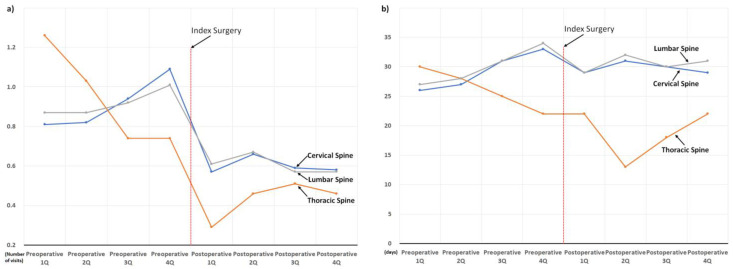
Perioperative changes in the use of sleep medication and hospital visits owing to sleep disorder according to the spinal regions: (**a**) changes in the number of hospital visits, and (**b**) changes in the prescribed duration of sleep medication.

**Figure 7 jcm-11-07402-f007:**
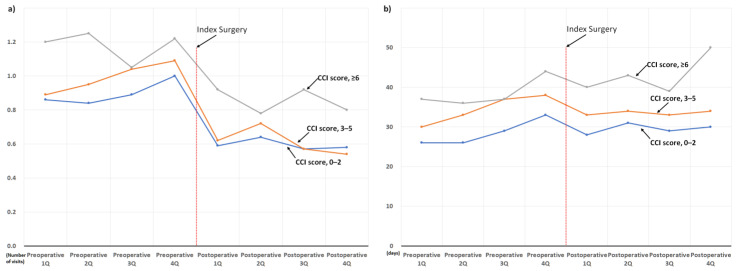
Perioperative changes in the use of sleep medication and hospital visits owing to sleep disorder according to the Charlson comorbidity index (CCI) score: (**a**) changes in the number of hospital visits, and (**b**) changes in the prescribed duration of sleep medication.

**Figure 8 jcm-11-07402-f008:**
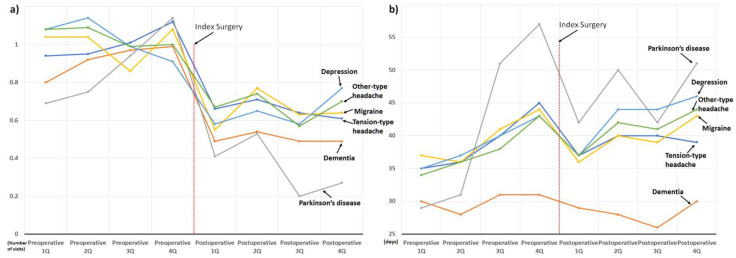
Perioperative changes in the use of sleep medication and hospital visits owing to sleep disorder according to the neuropsychiatric disorders: (**a**) changes in the number of hospital visits, and (**b**) changes in the prescribed duration of sleep medication.

**Figure 9 jcm-11-07402-f009:**
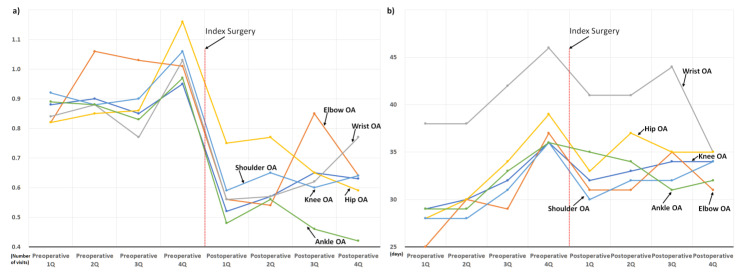
Perioperative changes in the use of sleep medication and hospital visits owing to sleep disorder according to the osteoarthritis of the extremities: (**a**) changes in the number of hospital visits, and (**b**) changes in the prescribed duration of sleep medication.

**Figure 10 jcm-11-07402-f010:**
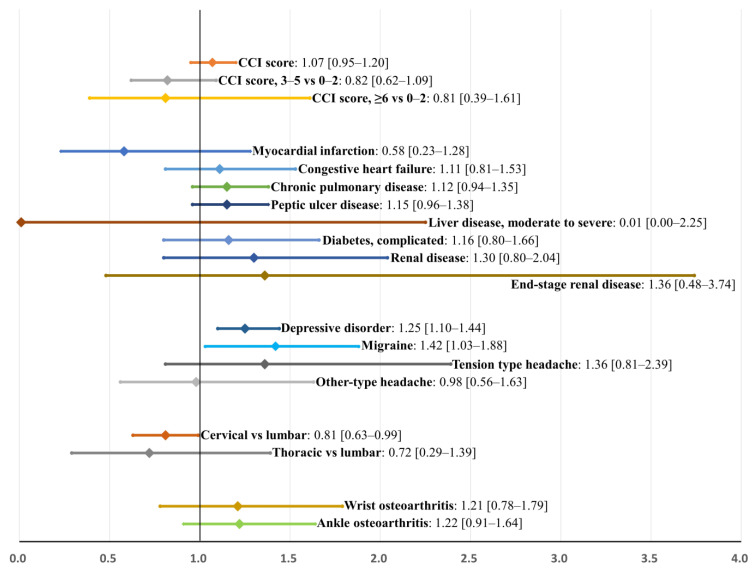
Factors associated with non-improvement in sleep medication after surgery: a multivariable analysis. Bootstrap-adjusted odds ratios and their 95% confidence intervals are presented.

**Table 1 jcm-11-07402-t001:** Baseline characteristics of the cohort patients.

Variables	Categories	All	Patients without Postoperative Improvement in Sleep Medication	Patients with Postoperative Improvement in Sleep Medication	Proportions of Patients without Improvement	Standardized Mean Difference
Number of patients		3183	776	2407	24.4%	
Age	Mean ± SD	66.7 ± 10.7	67.2 ± 10.5	66.5 ± 10.8		0.065
	20–49	213 (7)	50 (6)	163 (7)	23.5%	
	50–69	1525 (48)	366 (47)	1159 (48)	24.0%	
	70+	1445 (45)	360 (46)	1085 (45)	24.9%	
Sex	Men	1308 (41)	307 (40)	1001 (42)	23.5%	0.046
	Women	1875 (59)	469 (60)	1406 (58)	25.0%	
Region	Urban	2714 (85)	660 (85)	2054 (85)	24.3%	0.012
	Rural	469 (15)	116 (15)	353 (15)	24.7%	
Hospital	Tertiary	640 (20)	153 (20)	487 (20)	23.9%	0.062
	General	660 (21)	172 (22)	488 (20)	26.1%	
	Others	1883 (59)	451 (58)	1432 (59)	24.0%	
Spinal regions	Cervical	461 (14)	96 (12)	365 (15)	20.8%	0.134
	Thoracic	35 (1)	7 (1)	28 (1)	20.0%	
	Lumbar	2687 (84)	673 (87)	2014 (84)	25.0%	
Charlson comorbidity index score	Mean ± SD	1.61 ± 1.46	1.75 ± 1.52	1.56 ± 1.44		0.136
	0–2	2427 (76)	572 (74)	1855 (77)	23.6%	
	3–5	696 (22)	185 (24)	511 (21)	26.6%	
	≥6	60 (2)	19 (2)	41 (2)	31.7%	

**Table 2 jcm-11-07402-t002:** Comorbidities of the cohort patient.

Variables	Categories	All	Patients without Postoperative Improvement in Sleep Medication	Patients with Postoperative Improvement in Sleep Medication	Proportions of Patients without Improvement	Standardized Mean Difference
General comorbidities	Myocardial infarction	40 (1)	7 (1)	33 (1)	17.5%	0.233
	Congestive heart failure	159 (5)	45 (6)	114 (5)	28.3%	0.118
	Cerebrovascular disease	441 (14)	113 (15)	328 (14)	25.6%	0.043
	Peripheral vascular disease	531 (17)	140 (18)	391 (16)	26.4%	0.070
	Chronic pulmonary disease	1059 (33)	285 (37)	774 (32)	26.9%	0.112
	Rheumatologic disease	161 (5)	42 (5)	119 (5)	26.1%	0.053
	Peptic ulcer disease	745 (23)	205 (26)	540 (22)	27.5%	0.119
	Liver disease					
	Mild	273 (9)	70 (9)	203 (8)	25.6%	0.041
	Moderate to severe	5 (0)	1 (0)	4 (0)	20.0%	0.140
	Diabetes					
	Uncomplicated	828 (26)	206 (27)	622 (26)	24.9%	0.020
	Complicated	245 (8)	70 (9)	175 (7)	28.6%	0.129
	Hemiplegia or paraplegia	26 (1)	1 (0)	25 (1)	3.8%	1.156
	Renal disease	105 (3)	34 (4)	71 (3)	32.4%	0.226
	End-stage renal disease	16 (1)	6 (1)	10 (0)	37.5%	0.344
	Osteoporosis	609 (19)	144 (19)	465 (19)	23.6%	0.027
Concurrent neuropsychiatric disorders	Depressive disorder	1560 (49)	423 (55)	1137 (47)	27.1%	0.161
	Dementia	106 (3)	29 (4)	77 (3)	27.4%	0.089
	Parkinson disease	51 (2)	13 (2)	38 (2)	25.5%	0.033
	Migraine	202 (6)	67 (9)	135 (6)	33.2%	0.256
	Tension-type headache	180 (6)	59 (8)	121 (5)	32.8%	0.243
	Other-type headache	246 (8)	76 (10)	170 (7)	30.9%	0.197
Concurrent osteoarthritis of extremities	Shoulder	349 (11)	90 (12)	259 (11)	25.8%	0.047
	Elbow	78 (2)	18 (2)	60 (2)	23.1%	0.041
	Wrist	94 (3)	27 (3)	67 (3)	28.7%	0.127
	Hip	299 (9)	80 (10)	219 (9)	26.8%	0.076
	Knee	991 (31)	258 (33)	733 (30)	26.0%	0.071
	Ankle	189 (6)	53 (7)	136 (6)	28.0%	0.111

**Table 3 jcm-11-07402-t003:** Factors associated with poor improvement in the sleep medication after surgery: internal validation using bootstrap sampling.

Variables	Categories	Univariable		Model 2 (Fully Adjusted)		Model 3 (Bootstrap Validation after Fully Adjusted)	
		Odds Ratio (95% Confidence Interval)	*p*-Value	Adjusted Odds Ratio (95% Confidence Interval)	*p*-Value	Adjusted Odds Ratio (95% Confidence Interval)	Bias (%)
Charlson comorbidity index score		1.09 [1.03–1.15]	0.002	1.07 [0.93–1.22]	0.364	1.07 [0.95–1.20]	−6.7
	3–5 vs. 0–2	1.17 [1.77–2.02]	0.102	0.83 [0.59–1.17]	0.284	0.82 [0.62–1.09]	4.7
	≥6 vs. 0–2	1.50 [0.87–2.61]	0.148	0.81 [0.35–1.89]	0.624	0.81 [0.39–1.61]	2.4
Comorbidities	Myocardial infarction	0.66 [0.29–1.49]	0.312	0.64 [0.28–1.47]	0.291	0.58 [0.23–1.28]	22.5
	Congestive heart failure	1.24 [0.87–1.77]	0.238	1.12 [0.75–1.62]	0.607	1.11 [0.81–1.53]	−10.3
	Chronic pulmonary disease	1.23 [1.03–1.45]	0.019	1.13 [0.92–1.39]	0.259	1.12 [0.94–1.35]	−3.8
	Peptic ulcer disease	1.24 [1.03–1.50]	0.023	1.15 [0.92–1.43]	0.233	1.15 [0.96–1.38]	1.5
	Liver disease, moderate to severe	0.78 [0.09–6.95]	0.820	0.56 [0.06–5.38]	0.615	0.01 [0.00–2.25]	678.3
	Diabetes, complicated	1.27 [0.95–1.69]	0.112	1.16 [0.78–1.72]	0.478	1.16 [0.80–1.66]	−2.3
	Renal disease	1.51 [0.99–2.29]	0.054	1.29 [0.77–2.16]	0.338	1.30 [0.80–2.04]	2.2
	End-stage renal disease	1.87 [0.68–5.16]	0.227	1.37 [0.45–4.16]	0.578	1.36 [0.48–3.74]	−3.4
Comorbidities associated with neuropsychiatric disorders	Depressive disorder	1.34 [1.14–1.57]	<0.001	1.25 [1.06–1.48]	0.008	1.25 [1.10–1.44]	1.4
	Migraine	1.59 [1.17–2.16]	0.003	1.42 [1.04–1.94]	0.028	1.42 [1.03–1.88]	−0.6
	Tension-type headache	1.56 [1.13–2.15]	<0.001	1.28 [0.68–2.44]	0.445	1.36 [0.81–2.39]	23.1
	Other-type headache	1.43 [1.08–1.90]	0.014	1.04 [0.59–1.84]	0.889	0.98 [0.56–1.63]	−139.0
Surgical regions	Cervical vs. lumbar	0.79 [0.62–1.00]	0.052	0.82 [0.64–1.04]	0.100	0.81 [0.63–0.99]	8.9
	Thoracic vs. lumbar	0.75 [0.33–1.72]	0.495	0.77 [0.33–1.78]	0.539	0.72 [0.29–1.39]	26.9
Concurrent osteoarthritis	Wrist	1.26 [0.80–1.98]	0.320	1.21 [0.76–1.92]	0.426	1.21 [0.78–1.79]	0.3
	Ankle	1.22 [0.88–1.70]	0.227	1.22 [0.87–1.71]	0.244	1.22 [0.91–1.64]	1.2

## Data Availability

The datasets generated for the current study are not publicly available because of Data Protection Laws and Regulations in Republic of Korea; however, the analyzed results are available from the corresponding authors upon reasonable request.

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
