# Peer review of "Changes in Sleep Problems in Patients Who Underwent Surgical Treatment for Degenerative Spinal Disease with a Concurrent Sleep Disorder: A Nationwide Cohort Study in 3183 Patients during a Two-Year Perioperative Period"

_jcm, 2022, doi:10.3390/jcm11247402_

Round 1

Reviewer 1 Report

I think this is a valuable report on sleep disturbances in patients with spinal disease. I read it with great interest, but I have one question.

As a clinician, I was wondering how much the date of hospital visit and the duration of prescription really affect the status of sleep disorders. The length of visits and prescriptions are related to the insurance system and the medical system of each country. For example, in Japan, sleeping pills cannot be prescribed for more than 30 days (to prevent problems such as large doses). In addition, patients are often required to visit the outpatient clinic relatively frequently before surgery for preoperative examinations and surgical explanations. This may naturally lead to a shorter prescription period for sleeping pills.

We believe that the results of this study will be greatly influenced by the insurance system and the healthcare system of each country. What are your thoughts on this point?

Reviewer 2 Report

The manuscript is a cohort study aimed to investigate the perioperative changes in sleep problems in patients who underwent surgical treatment for degenerative spinal disease and had concurrent sleep disorder, and to identify factors associated with poor sleep improvement after surgery. This study included 3183 patients aged ≥ 19 years who underwent surgery for degenerative spinal disease and had concurrent sleep disorder. Perioperative changes in the two target outcomes, including the use of sleep medication and hospital visits due to sleep disorders, were precisely investigated according to factors known to be associated with sleep disturbance. Logistic regression analysis was performed to identify independent factors associated with poor improvement in terms of sleep medication after surgery. All the estimates were validated using bootstrap sampling. During the one-year preoperative period, the use of sleep medications and hospital visits due to sleep disorder continuously increased. However, they abruptly decreased early after surgical treatment, and throughout the one-year postoperative period, they remained lower than those in the late preoperative period. At one-year follow-up, 75.6% (2407 of 3183) of our cohort showed improvement in sleep medication after surgery. Multi- variable analysis identified that the two neuropsychiatric disorders, depressive disorder and mi- graine, were associated with poor outcomes in the use of sleep medication after surgery.

I read the article with interest, the title is well thought out and faithfully reflects the content of the study, but it would be appropriate to specify that the study is a cohort study.

A)   The abstract is sufficiently developed, and it is useful to frame the characteristics and purpose of the study, but a few concerns are present:

Comment 1: It would be appropriate to add some more information about your conclusion.

B)    In the introduction, the characteristics of sleep problems in patients who underwent surgical treatment for degenerative spinal disease have been accurately described.

Comment 2: It would be appropriate to add some more information about sleep problems in patients who underwent surgical treatment for degenerative spinal disease, adding an appropriate bibliographical reference

Comment 3: The Aim of this study is not clear enough in the abstract, compared to the introduction.

C)    The Materials and Methods have been adequately developed.

Comment 4: It would be appropriate to specify under which specialist the patients with sleep disorders were treated and the type of sleep medication used.

Comment 5: It would be appropriate to improve the tables and figures used, some are unclear.

D)   The discussion is adequately developed.

Comment 6: It might be useful to add the limitations of the study and what should be investigated in subsequent studies to improve your results.

Finally, English language editing is needed.

Nevertheless, some minor changes are needed to be considered suitable for publication.
